# Simultaneous Optimization of Process Operational and Material Parameters for a 2-Bed Adsorption Refrigeration Process

**Marc Scherle * and Ulrich Nieken**

Institute of Chemical Process Engineering, University of Stuttgart, Boeblinger Strasse 78, D-70199 Stuttgart, Germany; ulrich.nieken@icvt.uni-stuttgart.de

\* Correspondence: marc.scherle@icvt.uni-stuttgart.de

**Abstract:** In process engineering, optimization is usually carried out without the simultaneous consideration of material and process. This issue is addressed in the following contribution. A model-based optimization is presented to improve the performance of adsorption heat pumps. Optimization is carried out in two steps. First, we optimize the operational parameters, the cycle time, and the thickness of the adsorbent for a given adsorption material. In a second step we use a material model to predict heat and mass transfer and adsorption capacity from structural material parameters. This allows us to vary the structural material parameters and calculate the optimal operational parameters for each adsorbent. The two-step optimization thus identifies optimal material properties together with corresponding optimal operational parameters. As constraints, a minimum specific cooling power (SCP) and the passive mass of heat transfer pipes are used. The coefficient of performance (COP) is taken as the objective function. We exemplarily demonstrate the approach for a two-bed adsorption chiller, carbide-derived carbon as the adsorbent, methanol as the sorptive and boron-nitrate as additive to improve heat conductivity. The approach can be easily extended to multi-bed installations and more sophisticated material models.

**Keywords:** adsorption refrigeration; combined material and process optimization; carbide-derived carbon (CDC); material model; two-step; model-based approach

---

## 1. Introduction

Thermally driven adsorption chillers (ACH) are promising for sustainable cooling. The necessary energy input can be supplied by solar heat or industrial waste heat, reducing primary energy consumption. Adsorption heat pumps have no moving parts, enabling silent operation and increasing operational life span. In addition, commonly used adsorption pairs are environmental friendly, compared to halogenated refrigerants used in conventional compression cooling machines.

Over the last decades, numerous studies have focused on the improvement of adsorption cooling machines. Due to poor system performance, low efficiency and comparably high costs, commercial dissemination is still limited [1,2]. One of the reasons why the potential is not fully exploited is the separation of research studies concerning either adsorbent properties or process parameters. Adsorbents for heat pump applications should allow for a high sorption capacity and low heat and mass transport resistance. Due to the contradicting nature of these requirements, technical adsorbents are always a compromise. Depending on the structural properties of the porous material, heat and mass transport and volumetric capacity are strongly related. For process simulation, quantitative expressions for heat and mass transport as well as sorption capacity are a prerequisite. Thus, a material model that connects the structure of the porous adsorbent to effective material properties is needed.

Another difficulty is the cyclic nature of adsorption processes, which requires numerically expensive, transient simulations of the cyclic steady state. So far, only a few studies have simultaneously analyzed material and process properties, trying to identify which structural material properties together with optimal operational parameters are promising for high energy efficiency.

For adsorption heat pump applications, simulations are usually used to gain insight into heat and mass transfer processes and to investigate different process designs. Pesaran et al. [3] give an extensive overview of numerical simulation studies of adsorption heat pumps. Their study revealed a lack of system-level studies, simultaneously taking adsorber bed geometry and operational parameters of the adsorption heat pump into account. The reviewed research studies can be categorized into three model types—thermodynamic models, lumped-parameter-models and distributed (heat and mass transfer) models. Sah et al. [4] give a review of modeling techniques using the same categorization, but further distinguish between different working pairs. It is pointed out that the major barriers of heat transfer inside the adsorber originate from the adsorbent material's low thermal conductivity, weak contact between heat exchanger surface and the adsorbent as well as the low heat transfer coefficients of the heat transfer fluid. It is emphasized that system design and adsorbent properties are mostly investigated separately.

Bau et al. [5] developed a dynamic optimization method to optimize adsorber-bed designs. By identifying optimal adsorption and desorption times, they calculated Pareto optimalities with respect to coefficient of performance (COP) and specific cooling power (SCP). They used a lumped parameter model to describe the adsorber bed behavior of different bed-designs, which need to be calibrated by experiments. Bau et al. justify the use of a lumped parameter model with the adsorbent bed's low thickness, allowing for pressure and temperature gradients to be neglected. Otherwise, the use of a distributed model is necessary. In a second and third step, they carry out optimization and design variations.

Miltkau et al. [6] studied the influence of zeolite layer thickness on dynamics and efficiency of a single-bed adsorption heat-pump with constant pressures in evaporator and condenser. A distributed parameter model was used, taking mass and heat transfer in the adsorbent layers into account. They discovered that a reduction of layer thickness leads to shorter cycle times and higher power densities. Dawoud et al. [7] as well as Schnabel and Füldner [8] used a similar approach to simulate the heat and mass transfer in a consolidated zeolite layer, aiming to optimize the zeolite structure with respect to layer thickness. In both investigations, the model was calibrated with volumetric adsorption measurements. Maggio et al. [9] used a predictive two-dimensional model of a two bed adsorption cooling machine, taking internal heat recovery into account. They identified the vapor permeability of the consolidated adsorbent bed, the wall heat transfer coefficient and the bed thickness as important parameters influencing the system performance. To decrease thermal resistance of adsorbent beds, one possibility is to use consolidated composites with heat additives such as expanded graphite [10]. Wang et al. [11] give an extensive review of applicable opportunities to improve the process performance. Besides a variety of different cycle setups, they present possibilities to enhance heat and mass transfer by either improving the adsorbent or the system design and the cycle mode.

Since the performance of an adsorption heat pump greatly depends on coupled heat and mass transfer rates inside the adsorber bed as well as process operational parameters, a simultaneous consideration is indispensable to gain an overall optimum. For each individual process setup, an optimum of cycle time, transport path for mass and energy, effective thermal conductivity and effective permeability exists. So far, few studies have concurrently dealt with process and material focused research. In the case that both aspects were considered, either simplified, lumped-parameter models were used [5,12], which are unable to represent mass and heat transport as a function of adsorbent thickness. Or an optimal adsorbent for specific process designs was selected out of a set of existing adsorbents [13].

In this work, we address the gap between material development and process optimization. For this purpose, we use a one-dimensional transport model along the characteristic dimension of the

adsorbent, in order to account for increasing transport resistance with adsorbent height. We focus on the well known two bed adsorption cooling process applying our optimization method in two steps. In a first step, we identify the optimum cycle time and material layer thickness for a given material. In a second step, the structure of the material is varied and the effective transport properties and overall sorption capacity is calculated using a material model. Then the first step is repeated with the new material.

Process efficiency in adsorption heat pumps is commonly characterized by the coefficient of performance (COP), which relates cooling power to energy input and specific cooling power (SCP), which in turn relates cooling power to adsorbent mass. Here, we use the COP under the constraint of a minimum required SCP as objective function. A second constraint is the unavoidable mass of the heat exchanger pipes, which is also an input parameter to process simulations.

The outline of this contribution is as follows: In Section 2.1 we give a brief introduction of the two-bed adsorption cooling process followed by the material model in Section 2.2. Section 2.3 provides the process model. The presentation of results in Section 3 starts with a brief discussion of a single cyclic simulation and is followed by finding optimal cycle times and material thickness for a given material. Finally, the structural parameters of the material are varied and a global optimum is identified.

## 2. Materials and Method

### 2.1. Adsorption Cooling Process

Heat transformation by adsorption heat pumps can be applied for heating or cooling purposes. In this work, we investigate the adsorption refrigeration process, but the methods can be transferred directly to adsorptive heating.

Adsorption processes are commonly of cyclic nature with alternating adsorption and desorption. Therefore, to ensure a quasi-steady cooling output, at least two adsorption beds are required. Schematically, the process set-up is shown in Figure 1, where bed 1 is connected to a condenser via valve V1 and is, thus, regenerated/desorbed and bed 2 is connected to an evaporator via valve V4. Figure 1a shows the heat inputs and outputs to the heat transfer fluids. The corresponding temperature levels of the heat exchanges are indicated in Figure 1b.

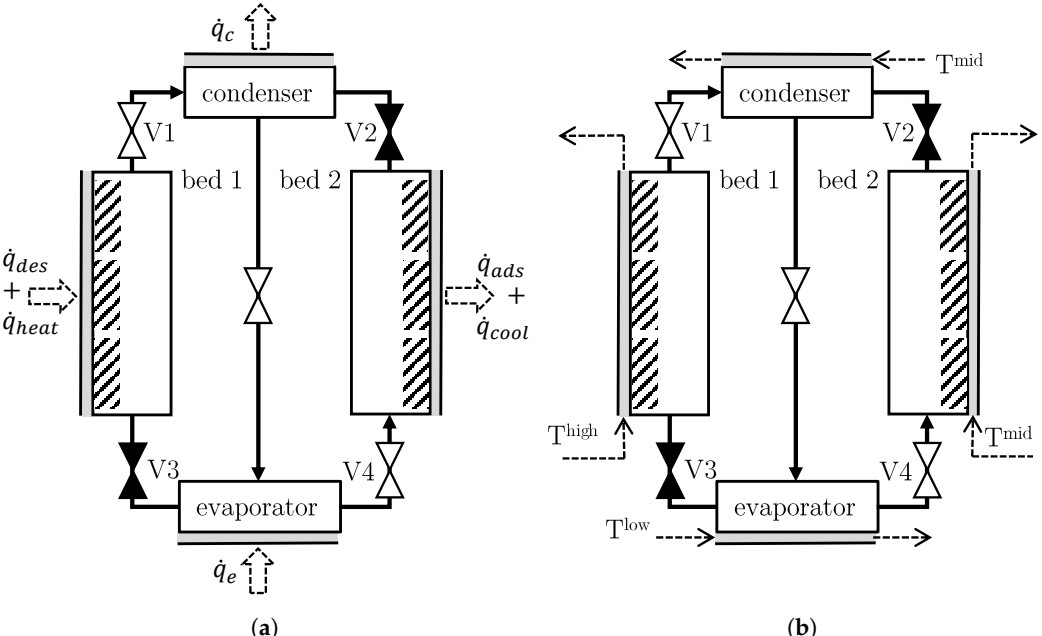

(**a**)　　　　　　　　　　　　　　　　　　　　(**b**)

**Figure 1.** Two bed adsorption refrigeration process with external heat fluxes. Bed 1 in desorption and bed 2 in adsorption. (**a**) External heat inputs and outputs. (**b**) Temperature levels.

Bed 1 is heated to a high temperature level (here $T^{high} = 80\,°C$), which lowers the adsorption potential of the adsorbent. The heat supply to the adsorbent during desorption is the energy input of the process. Desorbed refrigerant is liquefied in the condenser at ambient temperatures (here $T^{mid} = 30\,°C$). In the evaporator, the liquid refrigerant is evaporated by withdrawing energy at a low temperature level (here $T^{low} = 15\,°C$), which is the benefit of the process. The evaporated refrigerant is adsorbed in bed 2. The heat of adsorption is dissipated at ambient temperatures as well ($T^{mid}$).

After a specified time (here: half of the cycle time), valve 1 and 4 are closed. Bed 1 is cooled down, which initiates adsorption and leads to pressure decrease in the gas phase. Upon exceeding the evaporator pressure, valve 3 opens and vaporous refrigerant enters into bed 1. At the same time, bed 2 is heated up for regeneration. The pressure in the gas phase increases until condenser pressure is reached, which triggers the opening of valve 2 and bed 2 starts to regenerate. After a specified time, valve 2 and 3 are closed. Bed 1 is heated up again until the gas phase pressure reaches condenser pressure and bed 2 is cooled down until the pressure of the evaporator is reached. Subsequently, valve 1 and 4 are opened and the initial state in Figure 1a is restored and the cycle is completed.

The coefficient of performance (COP) and specific cooling power (SCP) are commonly used to rate process efficiency. The COP is the cooling output ($q_e$) in relation to energy input required for desorption and heating ($q_{des} + q_{heat}$). Here, $q_{heat}$ describes the heat provided to heat up the adsorbent bed from $T^{mid}$ to $T^{high}$. $q_{des}$ is the heat needed to desorb the refrigerant. This distinction is not necessary but is helpful in Section 2.5, where the calculation of the ideal COP is lined out. The SCP is cooling rate per specific mass of adsorbent compound ($m_{comp}$)

$$COP = \frac{\int_0^{t_{cycle}} \dot{q}_e \, dt}{\int_0^{t_{cycle}} (\dot{q}_{des} + \dot{q}_{heat}) \, dt} = \frac{q_e}{q_{des} + q_{heat}}, \tag{1}$$

$$SCP = \frac{\int_0^{t_{cycle}} \dot{q}_e \, dt}{m_{comp}} = \frac{q_e}{m_{comp}}. \tag{2}$$

## 2.2. Composite Plates of Carbide-Derived Carbon and Heat Additive

In the present work, we chose cuboid plates as model geometry of the adsorbent compounds with the advantage of a simple structure for material variation. In a technical application, cuboid plates can be mounted easily on flat heat exchanger tubes, which are rinsed with a fluid. The plates are composed of carbide-derived carbon (CDC), heat additive (ha) and (macro-) pores (v) in between (Figure 2).

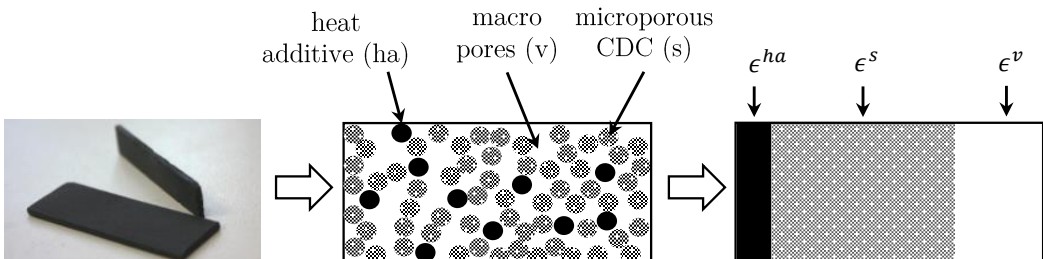

**Figure 2.** Image of manufactured plate compound (left side), sketch of the material composition consisting of solid heat additive, micro-porous adsorbent (CDC = carbide derived carbon) and transport (macro-) pores and simplified material model (right side).

CDC primary particles ($d^{eff}$=75–150 μm) are synthesized by chlorination of titanium carbide particles, showing extraordinary performance in the application of adsorption. Synthesis of CDC has two main technical advantages: The overall shape of the carbide is retained during chlorination and the pore size distribution can be adjusted by the choice of the precursor carbide and chlorination temperature [14]. Adsorption capacity as well as effective mass and heat transport of different plate compounds were investigated experimentally revealing the following interrelations—whereas heat

conductivity slightly increases by disorderly adding heat additive to the compound, the mass transfer slightly decreases. This behavior was already investigated by Jin et al. [15]. Additionally, the plate compound mass-related adsorption capacity linearly decreases with a linear addition of heat additive. Furthermore, the binder did not influence the adsorption capacity, which was also investigated by El-Sharkawy et al. [10]. These observations motivated the material model introduced in Section 2.2.2.

### 2.2.1. Physical Properties of Composite Adsorbent Plates

In the present work, we use methanol as sorptive. Adsorption equilibrium was modeled using the Dubinin-Astakhov adsorption isotherm

$$X_{eq}(p^v, T) = w_0 \varrho^{ad} exp\left[-\left(\frac{RTln\left(p^{sat}/p^v\right)}{E}\right)^n\right]. \tag{3}$$

This approach is well suited for micro-porous adsorbents where the idea of layer by layer surface adsorption loses its physical meaning and the idea of volume filling of micro-pores is decisive [16]. The model fitting parameters $w_0$ and $E$ describe the maximum specific pore volume and characteristic specific energy and the exponent $n$ accounts for homogeneity of the adsorbent respectively.

By using comparably small primary CDC-particles resulting in macro-pores in the range of a few µm, and CDC-plate thicknesses of up to 9 mm (in the simulations), macro-pore transport reveals the main mass transfer resistance. Due to operating pressures of around 50–200 mbar, using pure methanol as adsorbate [17], the mean free path of sorptive methanol molecules is in the range of the effective macro-pore diameter. The resulting Knudsen numbers are in the transition region, thus it is necessary to take viscous flow and Knudsen diffusion into account. Using the Dusty-Gas Model [18,19], an effective mass transfer coefficient in $\left[m_{comp}^2/s\right]$ is calculated

$$D^{\text{eff}} = \underbrace{\frac{\varepsilon^v D^{Kn}}{\tau}}_{D^{\text{Kn,eff}}(T)} + \frac{B_0^{\text{eff}}}{\eta}p, \tag{4}$$

with an effective permeability $B_0^{\text{eff}}$ for a porous composite material, $\tau$ known as tortuosity, $\varepsilon^v$ the bulk- or macro-porosity and $\eta$ the dynamic viscosity. Poiseuille introduced a formulation for isothermal, viscous flow through cylindrical pores with the permeability $B_0^{\text{eff}}$ [19]

$$B_0^{\text{eff}} = \frac{d_{\text{macro}}^2}{32}, \tag{5}$$

with the cylindrical macro-pore diameter $d_{\text{macro}}$. The effective Knudsen diffusion coefficient is temperature dependent [18]. Extracting the temperature independent part is convenient if experimentally investigated parameters are used in simulations with continuously varying temperatures

$$D^{\text{Kn,eff}}(T) = \frac{\varepsilon^v}{\tau}\frac{d_{\text{macro}}}{3}\sqrt{\frac{8R}{\pi MW}}\sqrt{T} = D^{\text{Kn,const}}\sqrt{T}. \tag{6}$$

By using an effective mass transfer approach with the spatial sorptive ($v$) density gradient across the layer thickness (coordinate $z$) as driving force, local mass flow densities can be expressed by

$$\dot{m}^v = -\frac{D^{\text{eff}}}{\varepsilon^v}\nabla\varrho^v = \frac{\dot{m}}{\varepsilon^v}. \tag{7}$$

Because comparably small CDC-primary particles are used, the intra-particle mass transport and, therefore, the mass transport in the micro-pores of the CDC primary particles is not limiting. Nevertheless, intra-particle resistance can be taken into account by the effective linear driving force

approach following Glueckauf [20], assuming Knudsen diffusion being the dominant transport phenomena in the micro-pores

$$k_{LDF} = \frac{15 D^{Kn}_{micro}}{r^2_{particle}} = \frac{15 d_{micro}/3 \sqrt{\frac{8RT}{\pi MW}}}{r^2_{particle}}. \tag{8}$$

Heat conduction along the main transport path of the composite plates was modeled with Fourier's law using effective heat conductivity

$$\dot{q} = -\lambda^{eff} \frac{\partial T}{\partial z}. \tag{9}$$

This heat flux density is related to the cross section area of the composite plates in $\left[W/m^2_{comp}\right]$, to which the phases contribute according to their respective volume fraction (compare Figure 2, right).

A combined heat conductivity for adsorbate and adsorbent was used, taking the local, specific uptake $X(z)$ of the CDC-primary particles into account, with $\lambda^{ad}$ the heat conductivity of adsorbed (liquid) methanol

$$\lambda^{s+ad} = \lambda^s \frac{1}{1+X} + \lambda^{ad} \frac{X}{1+X}. \tag{10}$$

To further consider plates with added particles of high thermal conductivity (heat additive), a linear contribution of the heat additive was used to calculate the overall heat conductivity of heat additive, CDC and adsorbate

$$\frac{\lambda^{eff}}{1-\varepsilon^v} = \lambda^{ha+s+ad} = \frac{\varepsilon^{ha}}{\varepsilon^{s+ad}+\varepsilon^{ha}} \lambda^{ha} + \frac{\varepsilon^{s+ad}}{\varepsilon^{s+ad}+\varepsilon^{ha}} \lambda^{s+ad}. \tag{11}$$

**Table 1.** Material parameters of an experimentally characterized, dry (without methanol adsorbed) CDC-plate with negligible binder content, without heat additive and physical properties of heat additive.

| | |
|---|---|
| $c_p^{eff,exp}$ $\left[J/kg_{comp}K\right]$ | 910 |
| $\lambda^{eff,exp}$ $\left[W/mK\right]$ | 0.563 |
| $D^{Kn,const,exp}$ $\left[m^2/\sqrt{K}s\right]$ | $4.04 \times 10^{-6}$ |
| $B_0^{eff,exp}$ $\left[m^2\right]$ | $1.32 \times 10^{-13}$ |
| $w_0^{exp}$ $\left[m^3/kg_{comp}\right]$ | $0.597 \times 10^{-3}$ |
| $E$ $\left[J/kg\right]$ | $207.4 \times 10^3$ |
| $n$ $\left[-\right]$ | 2.2 |
| $\varepsilon^{v,exp}$ $\left[m^3_{macro\,pores}/m^3_{comp}\right]$ | 0.37 |
| $\varrho^{bulk,exp}$ $\left[kg_{comp}/m^3_{comp}\right]$ | 602 |
| $c_p^{ha}$ $\left[J/kgK\right]$ | 1470 |
| $\lambda^{ha}$ $\left[W/mK\right]$ | 27 |
| $\varrho^{ha}$ $\left[kg/m^3\right]$ | 1900 |

The specific heat capacity was obtained by differential scanning calorimetry (DFC). $c_p^{eff}$ was further used to calculate the thermal heat conductivity from thermal diffusivity, which was obtained by a laser flash analysis. Mass transfer in the porous material was investigated by performing single component permeation experiments with a Wicke-Kallenbach cell [21], obtaining effective permeability and Knudsen diffusivity. Dubinin-Astakhov adsorption isotherms [16] are characterized experimentally with the static-volumetric method [22]. Pore size distribution, macro porosity and bulk density were obtained by mercury intrusion porosimetry.

In Table 1, material properties of an experimentally characterized composite plate with a negligible binder content are listed together with properties of heat addivtive, here boron nitrade. We found a linear dependence of effective heat conductivity with increasing amount of heat additive, but the dependence is much smaller in comparison with the thermal conductivity of the pure substance (up to 500 W/m/K for boron nitride [23]). This is due to non-percolating conditions of the heat conducting particles.

### 2.2.2. Material Model of Plate Compounds

We have developed a simple material model which reflects the experimental observations explained in Section 2.2 quite well. A sketch of the material model is depicted in Figure 2 on the right side. In this material model, the effective material properties are a superpositon of phase fractions (adsorbent ($\epsilon^s$), heat additive ($\epsilon^{ha}$) and macro-pores ($\epsilon^v$))

$$\sum_{j=1}^{3} \epsilon^j = 1 \quad \text{with j : ha, s, v.} \tag{12}$$

Mass transport (effective permeability) can be assumed to be in square proportion to the macro-porosity taking Hagen-Poiseuilles correlation (Equation (5)) into account and becomes zero when the macro-porosity is zero. Adsorption capacity and effective permeability were calculated using the following relations

$$w_0 = w_0^{exp} \frac{\varrho^s \varepsilon^s}{\varrho^s \varepsilon^s + \varrho^{ha} \varepsilon^{ha}} , \tag{13}$$

$$B_0^{\text{eff}} \sim (\varepsilon^v)^2 . \tag{14}$$

The effective heat conductivity was calculated using Equation (11). Furthermore, the following relations were used to determine bulk density and Knudsen diffusivity in dependence of the volume fractions

$$\varrho^{\text{bulk}} = \frac{M^s + M^{ha}}{V_{comp}} = \frac{m_{comp}}{s_{CDC}} = \varrho^{\text{bulk,exp}} \varepsilon^s + \varrho^{ha} \varepsilon^{ha}, \tag{15}$$

$$D^{\text{Kn,const}} = D^{\text{Kn,const,exp}} + \frac{D^{\text{Kn,const,exp}}}{\varepsilon^{v,\text{exp}}} (\varepsilon^v - \varepsilon^{v,\text{exp}}). \tag{16}$$

Using these correlations, the material space concerning effective permeability, effective heat conductivity and adsorption capacity for dry (no methanol adsorbed) adsorbent-compounds are shown in Figure 3.

At this point, it should be emphasized that the material parameters are competitive. For example, a high heat conductivity can only be reached by adding heat additive to the plate compound, which is at the expense of adsorption capacity referred to the adsorbent-compound mass. This effect becomes even greater, the larger the macro-pore volume gets, since the ratio of $\epsilon^{ha}/\epsilon^s$ increases.

With this material model, a varying composition of the composite CDC-plates can be represented. For example, by adding heat additive, the mass-specific adsorption capacity of the composite plate decreases as shown in Figure 3b.

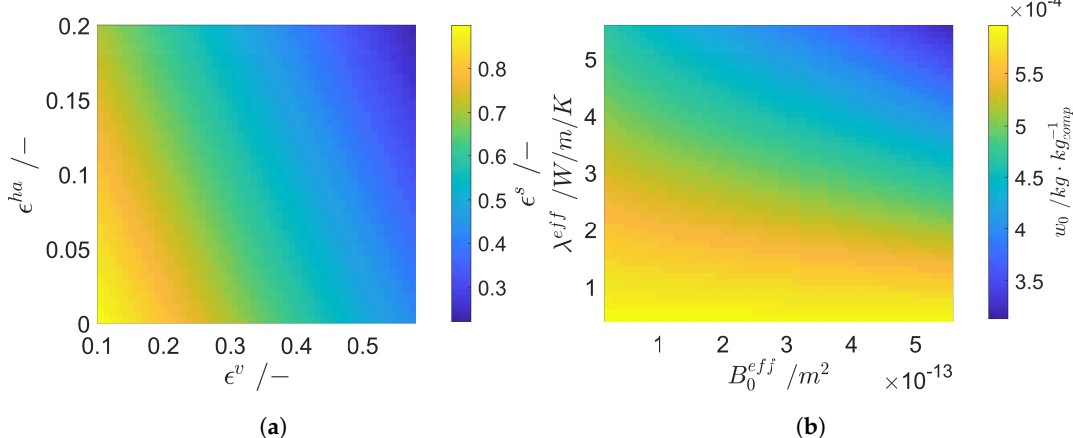

**Figure 3.** Parameter space of dry (no methanol adsorbed) adsorbent-compounds calculated with our material model. (**a**) Material space represented with the three volumetric proportions. (**b**) Material space with corresponding transport parameters and adsorption capacity.

### 2.3. Spatially Distributed Model of Composite Adsorbent Plates

A mathematical model was derived that neglects transport resistances caused by phase change in evaporator and condenser as well as by valves in interconnecting pipes. The resistance between heat source/sink and adsorbent material is neglected to exclusively investigate the transport resistances inside the composite CDC-plates. Gas holdup is neglected. Condenser and evaporator are modeled as ideal stirred tank reactors with a specified temperature and corresponding saturation pressure. Adsorption, mass- and heat transport in the adsorbent plates presented in Section 2.2 are modeled with a distributed, one dimensional parameter model to account for increasing transport resistances with increasing layer thickness. Transport over the side surfaces of the composite plates is neglected. The mass and heat flows from and to the CDC-plate are visualized in Figure 4.

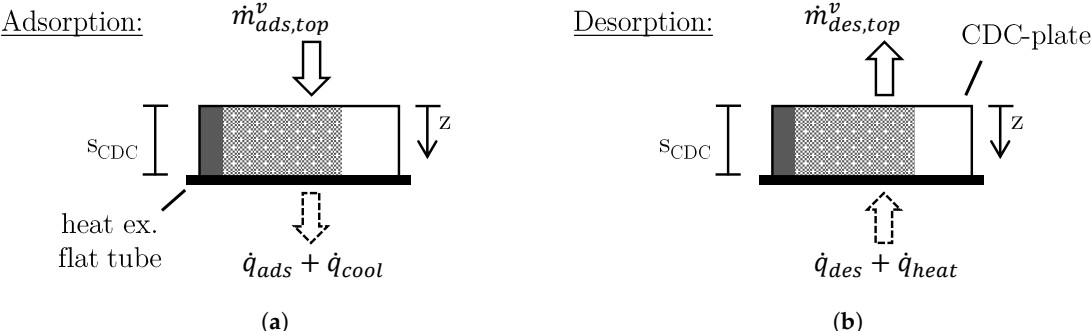

**Figure 4.** CDC plate on heat exchange flat tube with mass and energy flow densities during adsorption (**a**) and desorption (**b**).

The derivation of the spatially one-dimensional mass and energy balances is given in detail in Appendix A. The resulting balance equations are (see abbreviations for explanation of symbols)

$$\varepsilon^v \frac{\partial \varrho^v}{\partial t} + \varrho^{bulk}\frac{\partial X}{\partial t} + \varepsilon^v \frac{\partial \dot{m}_z^v}{\partial z} = 0, \tag{17}$$

$$\left( \varrho^{bulk} X c_p^{ad} + \varepsilon^{ha}\varrho^{ha}c_p^{ha} + \varepsilon^s\varrho^s c_p^s + \varepsilon^v\varrho^v(c_p^v - R/MW) \right)\frac{\partial T}{\partial t} - \varepsilon^v\, R/MW\, T\frac{\partial \varrho^v}{\partial t}$$
$$+ \varepsilon^v \dot{m}_z^v c_p^v \frac{\partial T}{\partial z} - \varepsilon^v v_z^v \frac{\partial p^v}{\partial z} + (1-\varepsilon^v)\frac{\partial \dot{q}_z^{ha+s+ad}}{\partial z} + \Delta_{ads}h\, \varrho^{bulk}\frac{\partial X}{\partial t} = 0. \tag{18}$$

Temperature changes caused by exothermic adsorption and endothermic desorption are taken into account by calculating enthalpy differences between adsorptive (v) and adsorbate (ad). The isosteric heat of adsorption includes condensation enthalpy and bonding energy [24]

$$\Delta_{ads}h = h^{ad} - h^v = (h^{ad} - h^l) + (h^l - h^v)$$
$$= \Delta^{bond}h + \Delta^{cond}h = \Delta^{cond}h + RTln\frac{p^{sat}(T)}{p^v}. \tag{19}$$

The upper index (l) accounts for a fictitious liquid phase being in thermodynamic equilibrium with the sorptive.

Micro-pore and adsorption kinetic is modeled with a linear driving force approach according to Glueckauf [20]. The transport coefficient is calculated using Equation (8)

$$\frac{\partial X}{\partial t} = k_{LDF}\left(X_{eq}(T,p) - X\right). \tag{20}$$

Heat conduction in the pores and mass transfer in solid adsorbent and adsorbate phase (e.g., surface diffusion) is neglected in this spatial one dimensional model.

To solve the set of partial differential Equations (17) and (18), a set of boundary conditions for the upper and lower layer of the CDC-plates is required. For the bottom layer ($z = s_{CDC}$), no-flux and Neumann boundary-conditions are applied for mass and energy balances respectively

$$\dot{m}^v\big|_{z=s_{CDC}} = 0, \tag{21}$$

$$\frac{\partial T}{\partial z}\bigg|_{z=s_{CDC}} = \frac{k^{pm}}{\lambda_{\text{eff}}}(T^{pm} - T|_{z=s_{CDC}}). \tag{22}$$

Here, $k^{pm}$ includes the heat transfer coefficient of plate compound to the passive mass of the heat exchanger wall (pm) and the heat conductivity of the heat exchanger wall. $k^{pm}$ is chosen to not be kinetically limiting.

A distinction between adsorption and desorption is necessary for the top layer boundary conditions at $z = 0$. Furthermore, comparing the pressure gradients between adsorber and evaporator as well as desorber and condenser is necessary to consider if the connecting valve is open or closed. In the case of adsorption, before and while the bed is connected to the evaporator, the following density Neumann and Dirichlet boundary conditions and temperature Neumann and Danckwerts [25] boundary conditions are applied

$$\frac{\partial \varrho^v}{\partial z}\bigg|_{z=0} = 0 \qquad\qquad \text{if } p_e^{sat} < p^v|_{z=0}, \tag{23}$$

$$\varrho^v|_{z=0} = \frac{p_e^{sat}MW}{T^v|_{z=0}R} \qquad\qquad \text{if } p_e^{sat} \geq p^v|_{z=0}, \tag{24}$$

$$\frac{\partial T}{\partial z}\bigg|_{z=0} = 0 \qquad\qquad \text{if } p_e^{sat} < p^v|_{z=0}, \tag{25}$$

$$\dot{m}|_{z=0}\, c_{p,e}\left(T|_{z=0} - T_e\right)$$
$$= \left(\dot{m}c_p^v\frac{\partial T}{\partial z} + \frac{\partial \dot{q}}{\partial z}\right)\bigg|_{z=0} \qquad \text{if } p_e^{sat} \geq p^v|_{z=0}. \tag{26}$$

In the case of desorption, before and while the bed is connected to the condenser, the following density and temperature boundary conditions are used

$$\left.\frac{\partial \varrho^v}{\partial z}\right|_{z=0} = 0 \qquad\qquad \text{if } p_c^{sat} > \left.p^v\right|_{z=0}, \qquad\qquad (27)$$

$$\left.\varrho^v\right|_{z=0} = \frac{p_c^{sat} MW}{\left.T^v\right|_{z=0} R} \qquad\qquad \text{if } p_c^{sat} \le \left.p^v\right|_{z=0}, \qquad\qquad (28)$$

$$\left.\frac{\partial T}{\partial z}\right|_{z=0} = 0. \qquad\qquad\qquad\qquad\qquad\qquad (29)$$

To complete the mathematical adsorber model, we use a lumped energy balance accounting for heating and cooling of the heat exchanger flat tubes

$$m^{pm} c_p^{pm} \frac{dT^{pm}}{dt} = \dot{q}^{pm} - \dot{q}^{hf}. \qquad\qquad (30)$$

It is essential to consider this equation to account for inevitable heating and cooling of passive masses. The passive mass is an important parameter that effects the process efficiency. The thinner the layer thickness of plate compounds, the greater the influence of the passive mass. The heat flow densities

$$\dot{q}^{pm} = k^{pm}(T^{pm} - \left.T\right|_{z=s_{CDC}}), \qquad\qquad (31)$$

$$\dot{q}_{ads} + \dot{q}_{cool} = \dot{q}^{hf} = k^{hf}(T^{hf} - T^{pm}) \qquad\qquad \text{adsorption,} \qquad\qquad (32)$$

$$\dot{q}_{des} + \dot{q}_{heat} = \dot{q}^{hf} = k^{hf}(T^{hf} - T^{pm}) \qquad\qquad \text{desorption.} \qquad\qquad (33)$$

are modeled using a fixed boundary temperature in the heat transfer fluid ($hf$). As stated previously, the transfer parameters $k^{hf}$ and $k^{pm}$ are chosen big enough to not affect process kinetics.

The mass flow densities to condenser and from evaporator with respect to the cross section area of the CDC plates are the flow densities entering or leaving the CDC plates

$$\dot{m}_e = \dot{m}^v_{ads,top}, \qquad\qquad (34)$$

$$\dot{m}_c = \dot{m}^v_{des,top}. \qquad\qquad (35)$$

Temperature levels of condenser and evaporator are constant, customized conditions. The heat withdrawn by the evaporator is the benefit of the process which is reduced by the enthalpy necessary to cool down the refrigerant flowing from warm condenser to cold evaporator. In this idealized model, the benefit of the process directly scales with the mass flow of adsorbed refrigerant (and thus evaporated refrigerant in the evaporator). The heat density to be dissipated from the condenser ($\dot{q}_c$) is composed of heat of condensation and enthalpy flow from the desorber

$$\dot{q}_e = \dot{m}_e \Delta h^{lv}(T_e) - \dot{m}_e c_p(T_c)\left[T_c - T_e\right], \qquad\qquad (36)$$

$$\dot{q}_c = \dot{m}_c \Delta h^{lv}(T_c) + \dot{m}_c c_p(\left.T_{des}\right|_{z=0})\left[\left.T_{des}\right|_{z=0} - T_c\right]. \qquad\qquad (37)$$

## 2.4. Numerical Implementation and Simulation Tools

We use a finite volume formulation for the spatial discretization of the presented set of partial differential equations (PDE), reducing it to a set of differential algebraic equations (DAE). The resulting first order spatial derivatives are discretized using central differences. Due to steep spatial gradients in the lower layers (see Section 3.1), a comparatively large number of 120 grid points is required. A grid analysis showed this number of grid-points being a good compromise between computational effort and accuracy.

The initial value problem

$$\overline{\overline{B}}\, \overline{y}' = \overline{f}(t,y) \quad \text{with} \quad \overline{y}(t_0) = \overline{y_0} \tag{38}$$

was dynamically solved for the solution vector $\overline{y}$ by time integration using the ode15s solver from MATLAB for stiff problems. An absolute and relative solver accuracy of $1 \times 10^{-5}$ was chosen.

For the determination of the cyclic steady state, we used absolute and relative error formulations comparing the entries in the solution vector $\overline{y}$ of consecutive cycles $i-1$ and $i$

$$Err^{abs} = max\left(\left|\left(|\overline{y}_i| - |\overline{y}_{i-1}|\right)\right|\right) < 10^{-6}, \tag{39}$$

$$Err^{rel} = max\left(\frac{\left|\left(|\overline{y}_i| - |\overline{y}_{i-1}|\right)\right|}{|\overline{y}_i|}\right) < 10^{-3}. \tag{40}$$

The check valves between adsorber and evaporator as well as desorber and condenser trigger a discontinuity in the boundary conditions (Equations (23)–(29)) depending on the pressure. Discontinuities are numerically demanding for multi-step solvers (such as ode15s). In our code, we use event-functions to detect the state dependent switching points. At these points, the solver stops, the boundary condition switches and the solver is reinitialized.

### 2.5. Process Limited Thermodynamic Efficiency

To evaluate the maximal efficiency of a heat engine, Carnot-efficiency, which assumes isothermal heat transformation, is commonly used. In sorption machines, heat transfer takes places at changing temperature levels making the Carnot-efficiency an insufficient measure of efficiency, which was extensively discussed by Núñez [26]. He suggests the efficiency evaluation of sorption systems using the energy balance of the idealized cyclic process (see Figure 5) solely taking thermodynamics into account

$$COP^{sorp.\ cooling}_{theoretical} = \frac{q_e^{ideal}}{q_{heat}^{ideal} + q_{des}^{ideal}}. \tag{41}$$

Figure 5 shows a Clausius-Clapeyron diagram of a real and idealized cyclic process.

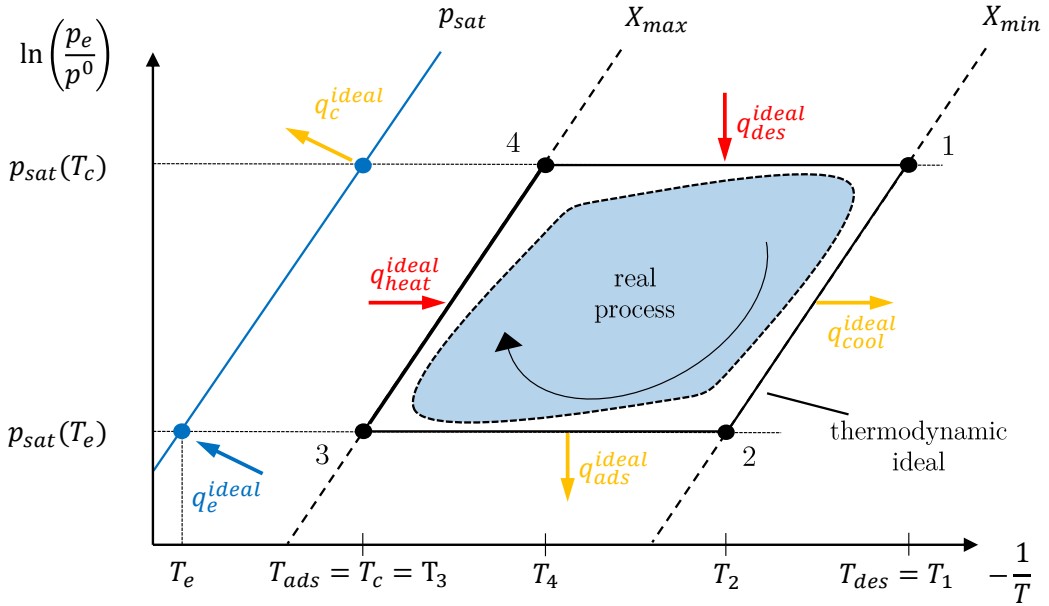

**Figure 5.** Thermodynamic ideal and real cyclic adsorption cooling process in a Clausius-Clapeyron diagram. The differences arise from kinetic transport resistances.

Deviation is caused by mass and heat transport resistances within the real process (in this work exclusively within the adsorbent compound). In the limiting case of infinite cycle time, the process *COP* (1) should always reach the theoretical $COP_{theoretical}^{sorp.\ cooling}$. Schematically, this can be identified by comparing the enclosing cross section areas of thermodynamically idealized and real processes. Reducing transport resistances will increase the enclosing cross section area of the real process until the limit of the thermodynamic process is reached.

The ideal cooling energy density is calculated with the difference of the equilibrium uptakes ($\Delta X_{eq}$) at the user specified temperature boundaries, heat of evaporation ($\Delta h^{lv}$) and the mass density of the adsorbent compound, reduced by the recycle of warm, liquid methanol from condenser to evaporator

$$q_e^{ideal} = m_{comp}\Delta X_{eq}\left(\Delta h^{lv} - c_{p,c}^l\left(T_c - T_e\right)\right). \tag{42}$$

The denominator of $COP_{theoretical}^{sorp.\ cooling}$ is separated into two terms

$$q_{heat}^{ideal} = \left[\left(mc_p\right)^{pm} + \left(mc_p\right)_{comp}\right]\Delta T_{31}$$
$$+ m_{comp}\left[X_{max}^{eq}c_p^{ad}\left(T_4 - T_3\right) + \frac{X_{max}^{eq} + X_{min}^{eq}}{2}c_p^{ad}\left(T_1 - T_4\right)\right], \tag{43}$$

$$q_{des}^{ideal} = m_{comp}\Delta X_{eq}\Delta_{ads}h. \tag{44}$$

The first term ($q_{heat}^{ideal}$) describes the heating of passive mass and adsorbent compound with constant heat capacity and adsorbate mass with varying heat capacity from $T_{ads}$ to $T_{des}$. The second term ($q_{des}^{ideal}$) describes the energy density required for desorption. This separation has been made to distinguish between energy needed to regenerate the adsorbent bed and heat required to heat up the bed between the temperature boundaries. At this point, the importance of the passive mass for a correct evaluation of the process must be emphasized. If the passive masses were not taken into account, an optimization would always result in a material with maximum transport parameters and minimum transport paths, since the adsorption capacity is no longer of importance. In the limit of long cycle times, the same COP would result regardless of the adsorbent mass used, which does not correspond to the real process. As we will show, this is oversimplified and passive mass should be accounted for.

Furthermore, heating of the adsorbate is divided into a term with constant specific uptake and a term with variable, averaged specific uptake.

By using the ideal heat density of evaporation $q_e^{ideal}$, a theoretical specific cooling power can be defined as well

$$SCP_{theoretical}^{sorp.\ cooling} = \frac{q_e^{ideal}}{m_{comp}t_{cycle}}. \tag{45}$$

### 2.6. Process Conditions

To reduce the free parameter space, we fixed some process conditions and the temperature levels. The values are listed in Table 2.

**Table 2.** Process boundary conditions and geometrical parameters kept constant for all simulations.

| | |
|---|---|
| $T^{high} = T^{des}$ [K] | 353.15 |
| $T^{mid} = T^{ads} = T^c$ [K] | 303.15 |
| $T^{low} = T^e$ [K] | 288.15 |
| $c_p^{pm}$ [J/kgK] | 500 |
| $m^{pm}$ [kg/m²] | 4 |

The importance of taking the passive mass into account was already discussed in Section 2.5. In this work, we use a specific mass related to the cross-sectional area of the CDC-plates to account for the passive mass. This parameter was calculated be assuming stainless steel flat tubes with a wall thickness of 0.5 mm with a density of 8000 kg/m$^3$. The formulation as a specific reference value was chosen in order to be able to scale the model to any cross-sectional area and, thus, cooling output.

## 3. Results and Discussion

In the following, we will first discuss the characteristic behavior of the process. Subsequently, we present a two step optimization method and discuss the results.

### 3.1. Simulation Results: Physical Process Description Based on Single Simulations

Simulations of the one-dimensional process model provide an insight into the spatial distribution of temperature and specific uptake over the thickness of the CDC-plates. In Figure 6 spatial temperature and specific uptake of one cycle in cyclic steady state are shown. The upper graphs show profiles during adsorption, the lower during desorption. In cyclic steady state, the initial profiles in the adsorption step are identical to the final profiles of the desorption step.

The initial temperature profile in the adsorption step is shown on the left side of Figure 6a. By switching the temperature boundary condition in the heat transfer fluid from $T^{high}$ to $T^{mid}$, a temperature drop on the lower layer of the CDC-plates (at 9 mm) occurs. During the adsorption step of 100 s (half of the cycle time), the temperature continuously decreases whereby the upper layers experience a smaller and significantly slower drop in temperature than the lower layers. This is not only due to transport resistance across the plate, but also by the exothermic adsorption, which preferably takes place on the upper side of the CDC-plates. Due to the high initial temperature, the pressure in the top layer exceeds the pressure of evaporator and the connecting valve stays closed (see Equation (23)). Rather, a relocation of refrigerant in the adsorbent takes place. When the pressure of the evaporator exceeds the adsorbent pressure, the connecting valve opens (see Equation (24)). This allows for a flux from the evaporator to the adsorber and, therefore, a withdrawal of energy at the evaporator temperature level.

The spatial profiles of the desorption step are shown in Figure 6b. After 100 s (second half of the cycle time), the desorption step starts by a change in the temperature boundary condition from $T^{mid}$ to $T^{high}$, which causes a temperature rise in the CDC-plates. Similar to the adsorption step, the adsorbent bed is not connected to the condenser directly due to the pressure condition for the valve opening (see Equation (27)). Until the pressure above the adsorbent reaches the condenser pressure (see Equation (28)), desorbing methanol from the lower layers is getting adsorbed in the middle layers. Once the connecting valve between adsorbent and condenser opens, the upper layers are preferentially regenerated over time.

Next, we briefly show the transient to the cyclic steady state. Figure 7 shows the **mean** specific uptake over several cycles for different combinations of cycle time and CDC-plate thickness.

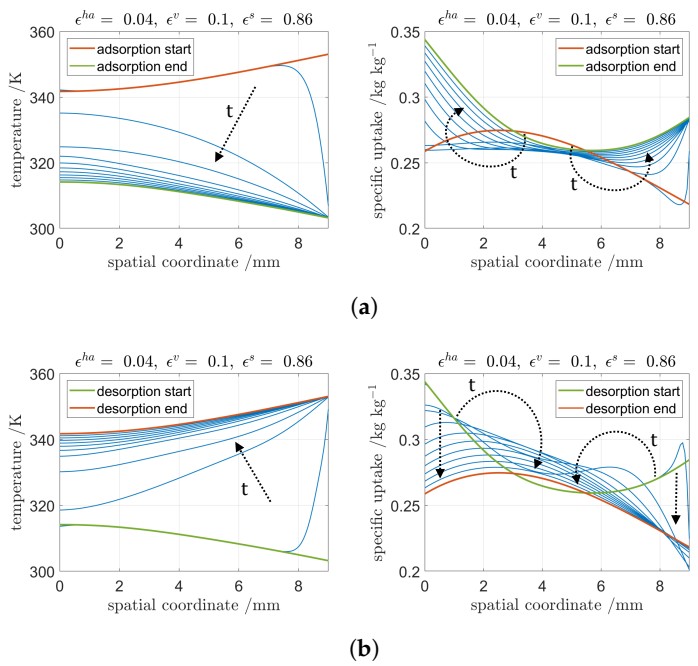

**Figure 6.** Spatial temperature and specific uptake profiles exemplary for a layer thickness ot 9 mm, cycle time of 200 s and a CDC-plate composition of $\epsilon^{ha} = 0.04$, $\epsilon^v = 0.1$ and $\epsilon^s = 0.86$. (**a**) Adsorption step: Spatial temperature and uptake profiles in the first half of the cycle time. (**b**) Desorption step: Spatial temperature and uptake profiles in the second half of the cycle time.

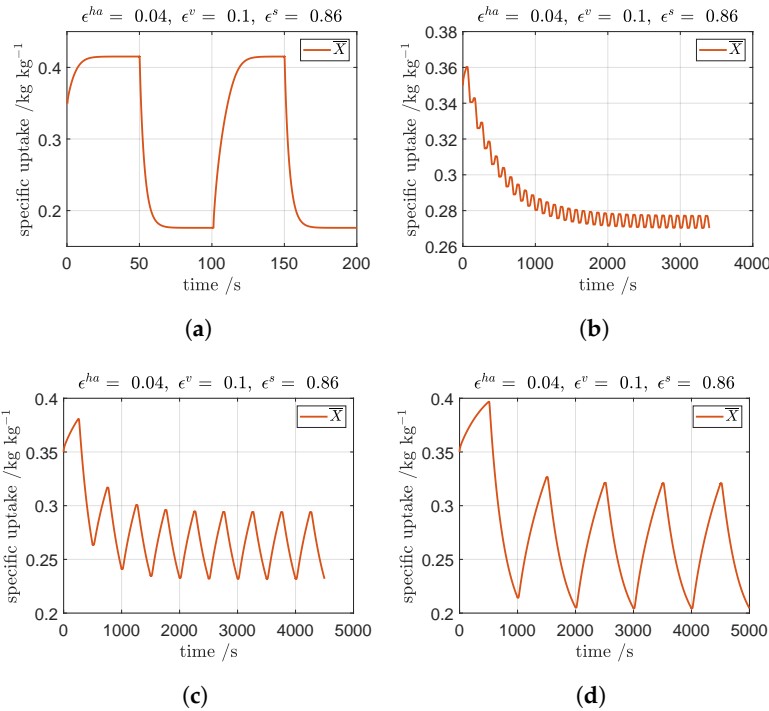

**Figure 7.** Mean specific uptakes from initialization to cyclic steady state for simulations with different layer thickness and cycle time. The composition of the CDC-plates is the same for these simulations with $\epsilon^{ha} = 0.04$, $\epsilon^v = 0.1$ and $\epsilon^s = 0.86$. (**a**) Cycle time of 100 s and layer thickness of 1 mm. (**b**) Cycle time of 100 s and layer thickness of 9 mm. (**c**) Cycle time of 500 s and layer thickness of 9 mm. (**d**) Cycle time of 1000 s and layer thickness of 9 mm.

When using thin CDC-plates (1 mm), the mean equilibrium uptake is reached in approximately 20 s after switching between adsorption/desorption and vice versa, which can be seen in Figure 7a. The difference between equilibrium uptake in adsorption and desorption is ca. 0.24 kg/kg. Therefore, cycle times longer than 40–50 s do not provide any improvement in terms of COP and come with a reduction in SCP. In contrast, when using thick plates (9 mm) and the same cycle time of 100 s, the thermodynamic equilibrium is not achieved, as can be seen in Figure 7b with a difference in specific uptake of ca. 0.07 kg/kg. Furthermore, when using the same starting values, the simulation with 9 mm plate thickness requires more cycles to reach cyclic steady state in comparison to the simulation with 1 mm plate thickness. The advantage of thicker plates (greater COP) comes into play when longer cycle times are used because the available adsorption capacity is better utilized. Figure 7c,d show mean specific uptakes of simulations for a plate thickness of 9 millimeter and cycle times of 500 s and 1000 s respectively. For these thick plates, even a cycle time of 1000 s is insufficient to reach equilibrium.

### 3.2. Objective Function and Two Step Optimization Approach

The workflow of the proposed two step optimization approach is shown in Figure 8.

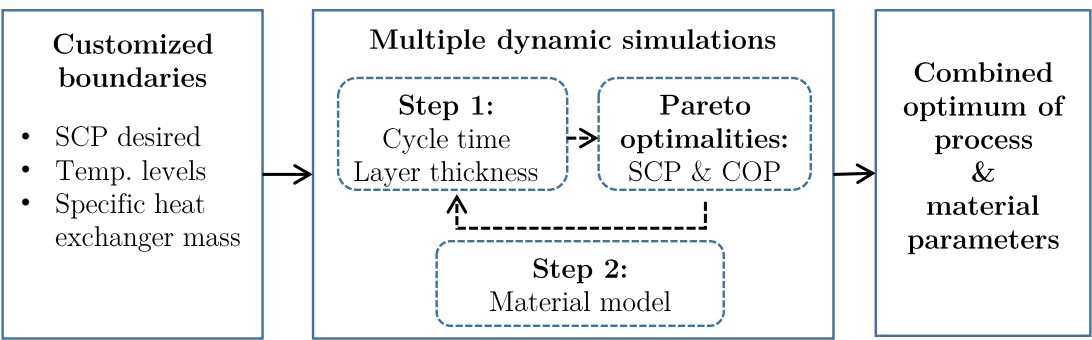

**Figure 8.** Workflow of two step process and material optimization approach.

The three temperature levels, desired SCP and the specific mass of the heat exchanger flat tubes have to be specified initially.

In the first step, cycle time ($t_{cycle}$) and adsorbent compound layer thickness ($s_{CDC}$) are varied in a wide range for a given material. For each unique combination, the spatially distributed dynamic model is solved for cyclic steady state. In a post processing step, a Pareto frontier is identified, regarding the two objectives COP and SCP. For every SCP specified, a pair of optimal cycle time and layer thickness corresponding to the maximal COP can be identified. If the invest price for the adsorbent is high and/or heating energy is freely available, the user would aim a high cooling power density. From this result a rather low optimal COP, thin layer thickness and short cycle time. On the other hand, if the energy efficiency is the main user objective, the process would be operated with a low SCP, large optimal layer thickness and long cycle times. Taking both, SCP and COP, into account is advantageous since the user can decide whether a high power density or efficiency is the main goal.

In the second step, the material model presented in Section 2.2.2 is used to vary the material parameters of the composite plates in a physical feasible range. For each combination of material and layer thickness, a Pareto frontier for SCP and COP is identified.

Finally, after setting up the parameter space with multiple dynamic, cyclic simulations, an optimal operational strategy with optimal material parameters can be identified. This optimum depends on the user specified objective concerning the desired $SCP^{desired}$.

Even though the presented method is demanding in terms of calculation effort, it offers some advantages compared to other methods. Using a one-dimensional model with effective transport parameters is advantageous. Once a material is parameterized, a variation of the layer thickness is possible without the need for new experiments and parameterizations. In comparison,

zero-dimensional models usually only using one kinetic parameter, must be parameterized for each new layer thickness. This requires a high effort in material production and experimental characterization. Once a valid material model has been found, the method can be applied to a variety of different process setups. Another advantage of this method is its flexibility. It can be easily extended to more sophisticated material models and applied to several different process setups.

Each combination of material and process results in a unique optimal curve. Visualizing the final optimization result in a master curve that contains all relevant material and process related optimization is another advantage of this method, because it allows for an easy comparison of different process arrangements and material models.

### 3.2.1. First Step-Process Parameters Optimization for One Material

In the first step, a parameter variation was carried out to find the maximum achievable $COP^{max}$ in the two-dimensional parameter space concerning process cycle time and layer thickness of CDC-plates. Cycle time was varied in a wide range to generate Pareto frontiers of COP and SCP. Figure 9 shows the results exemplary for a layer thickness of two and four millimeters for one material with volume fractions of 4% heat additive, 10% macro-pores and 86% of CDC.

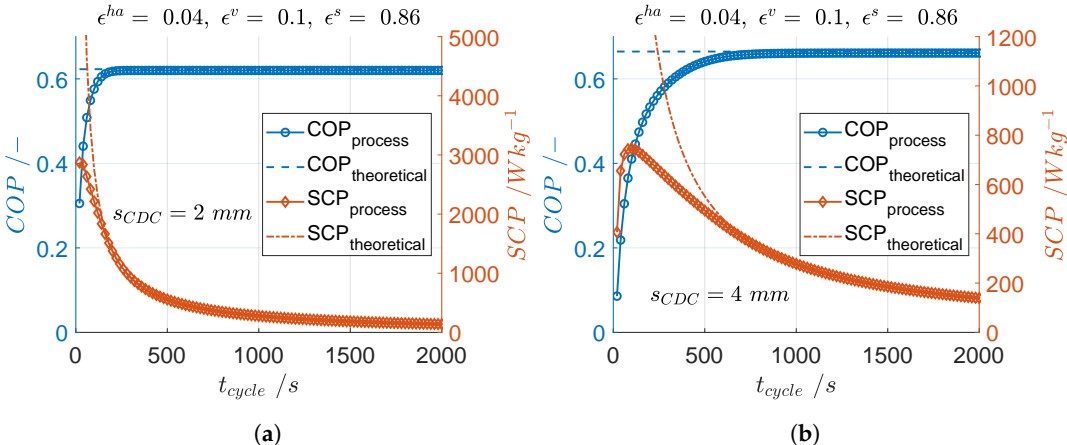

(**a**)                                        (**b**)

**Figure 9.** Coefficient of performance and specific cooling power for a wide variation of cycle time and one material with different layer thickness. (**a**) Optimization of cycle time for a constant CDC-plate layer thickness of 2 mm. (**b**) Optimization of cycle time for a constant CDC-plate layer thickness of 4 mm.

For long cycle times, the specific uptake reaches equilibrium uptake and, thus, the theoretical COP and SCP are reached. Whereas COP is monotonously rising with cycle time, SCP has an optimum and is declining with longer cycle times. By comparing Figure 9a,b, the effect of plate thickness becomes apparent. The usage of thicker CDC plates (Figure 9b vs. Figure 9a) results in a higher COP, as there is a more favorable ratio of active to passive masses. The disadvantage of thicker layers, however, is that the dynamics of the transport processes decrease due to longer transport distances. This means that the maximum SCP is significantly lower and reached at longer cycle times compared to thin layers. Similarly, the theoretical COP is reached at significantly longer cycle times when using thicker adsorbent compounds. This shows the trade off between energy efficiency and power density. When optimizing adsorption heat pump processes, both aspects are interdependent. This can be illustrated by plotting COP over SCP. A Pareto frontier can be identified, which is exemplary shown for a layer thickness of four millimeters in Figure 10.

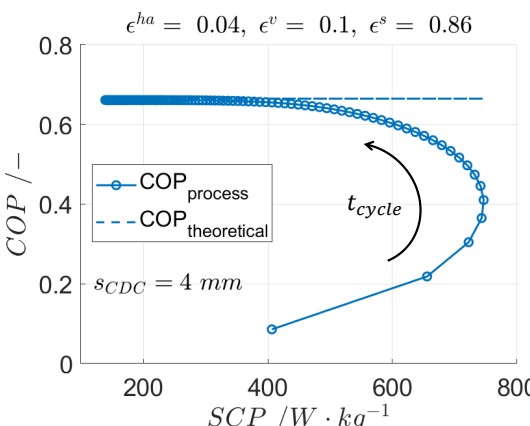

**Figure 10.** Pareto frontier of coefficient of performance (COP) and specific cooling power (SCP) for a constant CDC-plate layer thickness of four millimeter.

The optimization in favor of a large specific cooling capacity is always accompanied by a loss of energy efficiency. By repeating this procedure for different layer thicknesses of the composite plates from 1–9 mm, several Pareto frontiers have been calculated.

Using these Pareto frontiers, each optimal $COP^{opt}$ can be identified in dependence of the desired SCP for a given adsorbent. Exemplary, this is visualized in Figure 11 for a desired $SCP^{\text{desired}}$ of 500 W/kg.

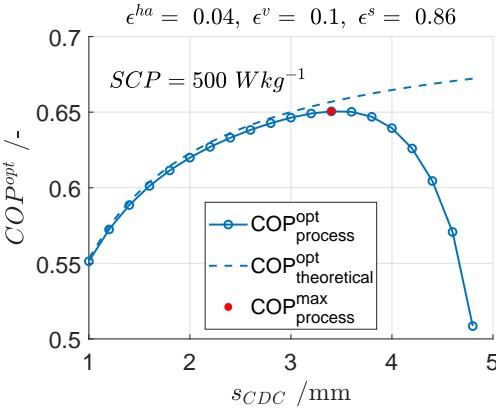

**Figure 11.** Maximum achievable $COP_{\text{process}}^{\max}$ in the two-dimensional parameter space concerning cycle time and CDC-plate layer thickness exemplary for a user desired $SCP^{\text{desired}}$ of 500 W/kg.

This analysis leads to a maximum achievable $COP_{\text{process}}^{\max}$ for a given material in the two-dimensional parameter space concerning cycle time and layer thickness. Further, the results in Figure 11 reveal the transport limitations of thick layers and the increasing divergence from equilibrium and, therefore, $COP_{\text{theoretical}}^{\text{opt}}$. In this example, the dynamics of plates thicker than 4.8 mm is too low to provide the desired SCP. The optimal layer thickness is 3.4 mm, the corresponding cycle time 547.4 s.

### 3.2.2. Second Step-Material Optimization

In the second step, we aimed for a combined optimization of process and material parameters to reach a global rather than a local optimum. Therefore, we extended the parameter space by two material parameters and optimized the COP in a four-dimensional parameter space with respect to a desired $SCP^{\text{desired}}$ of 500 W/kg.

By using the material model, we varied the volume fractions of heat additive and macro-pores of the CDC-plates and related the remaining material parameters as described in Section 2.2.2. In addition to kinetic parameters, structural parameters and the adsorption capacity are adapted to a changed composition of the CDC plates. Thus, in this material model, the competing behavior of effective

thermal conductivity, permeability and adsorption capacity is taken into account, which corresponds to the observations of the experimental material variation.

By repeating step one, different $COP_{process}^{max}$ can be identified for different materials from the material space presented in Figure 3. From the distribution of all $COP_{process}^{max}$, an optimum can be determined in the four-dimensional parameter space, which includes cycle time, layer thickness and volume fractions of macro-pores and heat conductivity additive.

Figure 12 exemplary shows a set of $COP_{process}^{max}$ for a variation of the macro-pore porosity from 0.1 to 0.58 with a constant volume fraction of heat additive. A unique maximum for $COP_{process}^{max}$ can be identified, providing the optimum void fraction of $\epsilon^v = 0.3$ for a fixed amount of heat additive ($\epsilon^{ha} = 0.04$). The corresponding optimal layer thickness for this material is 5.2 mm, the respective cycle time 531.7 s.

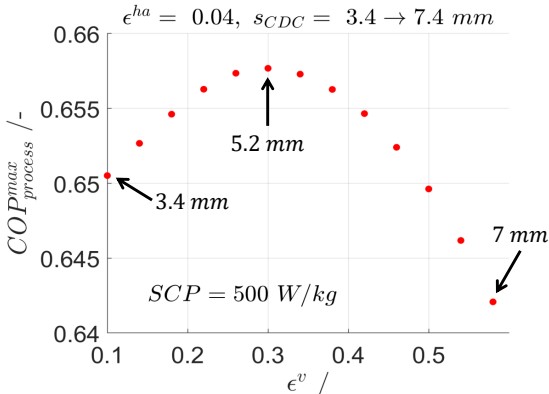

**Figure 12.** Varying, maximal $COP^{max}$ for different macro-pore volume fractions and layer heights.

By expanding this analysis and further varying the void fraction of heat additive, a global optimum in the four-dimensional parameter space can be identified.

The surface plot in Figure 13a reveals an optimal material composition with 30 vol% macro-pores, 4 vol% heat additive and 66 vol% adsorbent. The corresponding transport parameters can be identified from Figure 13b. Clear optimums of $COP_{process}^{max}$ can be identified, which are marked with a square. These points represent the global optimum in terms of cycle time, CDC-plate layer thickness and material composition.

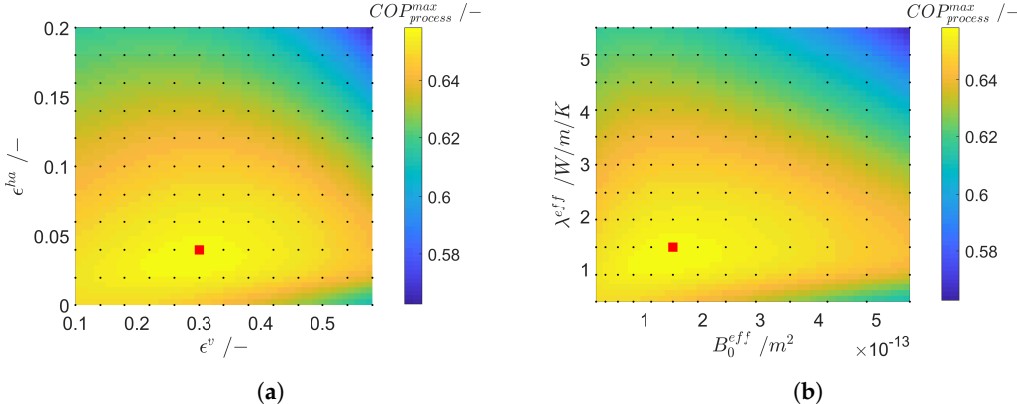

(**a**)                    (**b**)

**Figure 13.** Global optimum concerning process and material parameters. For each composition, cycle time and plate layer thickness are optimal. The squares represent the global optimum, where additionally the composition of the CDC-plate is optimal. (**a**) $COP_{process}^{max}$ with respect to volume ratios of heat additive, macro-pores and adsorbent. (**b**) $COP_{process}^{max}$ with respect to effecitve heat conductivity and permeability.

By using our material model with the competing material properties, a non-trivial, global optimum has been identified. Since the thermal conductivity of pure CDC primary particles is low, they present the greatest transport resistance. By adding heat additive, the effective thermal conductivity can be improved, making mass transfer the limiting factor. The increase of $\epsilon^{ha}$ leads to a decrease of the adsorption capacity. Therefore, there is an optimal fraction from which a further addition of thermal conductivity additive worsens the process performance.

### 3.3. Global Optimum: Optimal, Characteristic Curve for 2-Bed Adsorption Refrigeration Process

The presented two step optimization approach can be performed for any desired SCP within feasible limits. The corresponding $COP_{\text{process}}^{\text{max}}$ is shown in Figure 14 (left) and the corresponding cycle times, layer thickness and material compositions are listed in the table (on the right).

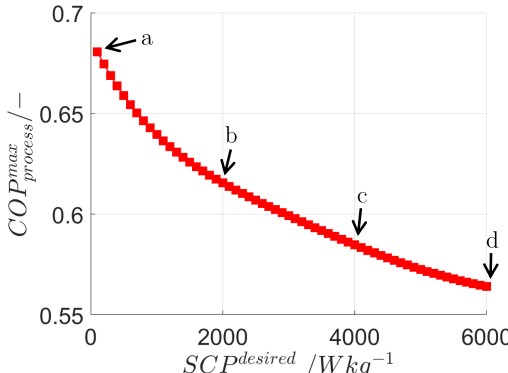

| | a | b | c | d |
|---|---|---|---|---|
| $t_{cycle}$ /$s$ | 2000 | 133.2 | 61.9 | 41.1 |
| $s_{CDC}$ /$mm$ | 8.4 | 2.6 | 2.2 | 1.8 |
| $\epsilon^{ha}$ /$-$ | 0.02 | 0.04 | 0.06 | 0.06 |
| $\epsilon^{v}$ /$-$ | 0.26 | 0.3 | 0.34 | 0.34 |
| $\epsilon^{s}$ /$-$ | 0.72 | 0.66 | 0.6 | 0.6 |

**Figure 14.** Optimal design layout for different desired values of SCP with corresponding optimal cycle time, layer thickness and material composition.

This analysis produces an optimal curve, which is characteristic for the examined 2-bed adsorption refrigeration process and the considered material model. As the desired SCP increases, the achievable COP decreases. The higher the desired SCP, the shorter the optimal cycle times, the thinner the plates and all the more macro-porosity and heat additive is required.

This analysis is very useful for comparing different process setups. A process setup with heat regeneration, for example, would produce a characteristic curve with larger, optimal COP's but with a steeper decline.

## 4. Summary and Conclusions

In this work, we developed a two-step method for the combined optimization of process and material related parameters. The method was exemplified for an adsorption refrigeration process using carbide derived carbon as adsorbent and methanol as sorptive. A process model for adsorption cooling devices was developed that counts for heat and mass transport limitations in one spatial direction. In addition, which was motivated by experimental investigations, a simple material model was introduced. However, more complex material models can easily be integrated in the workflow.

We showed that a global rather than a local optimum can be identified by using our approach, which combines process operational parameters and material structural properties. Accounting for the Pareto frontiers concerning SCP and COP, we used a combination of these evaluation criteria as cost function of our optimization. Finally, a characteristic curve was produced for the 2-bed adsorption refrigeration process with the underlying material model providing a basis for a comparison with other process setups.

However, the method is generally applicable and can be transferred to any target function and to other processes in the field of adsorption heat pumps. It allows a joint optimization of process and material parameters to obtain a global optimum for which the presented workflow

can be used. Numerical optimization methods may speed up the calculations if only the global optimum is of interest.

**Author Contributions:** Conceptualization, U.N.; methodology, M.S.; software, M.S.; validation, M.S.; formal analysis, M.S.; investigation, M.S.; resources, M.S. and U.N.; supervision, U.N. All authors have read and agreed to the published version of the manuscript.

**Funding:** This research was funded by the German Research Council grant number NI 932/10-1.

**Acknowledgments:** The authors gratefully acknowledge the funding of the German Research Council (DFG). The authors would like to acknowledge the provision of the samples by the working group of B.J.M Etzold, Technical University of Darmstadt. URL: www.etzoldlab.de.

**Conflicts of Interest:** The authors declare no conflict of interest.

## Abbreviations

The following abbreviations are used in this manuscript:

*Latin letters*

| | |
|---|---|
| $A$ | cross sectional area [$m^2$] |
| $B_0$ | permeability [$m^2$] |
| $cp$ | specific heat capacity [J/kg/K] |
| $D^{eff}$ | eff. transport coefficient [$m^2_{comp}/s$] |
| $d$ | diameter [m] |
| $E$ | charact. ads. energy [J/mol] |
| $h$ | specific enthalpy [J/kg] |
| $j$ | diffusive flux density [$kg/s/m^2$] |
| $k$ | heat transfer coeff. [W/m/K] |
| $k_{LDF}$ | linear driving force coeff. [$1/s$] |
| $M$ | mass [kg] |
| $m$ | (cross sectional area) specific mass [$kg/m^2$] |
| $\dot{m}$ | mass flow density [$kg/m^2/s$] |
| $MW$ | molecular weight [kg/mol] |
| $n$ | exponent Dusty-Gas-Model [$-$] |
| $p$ | pressure [Pa] |
| $q$ | heat density [$J/m^2$] |
| $\dot{q}$ | heat flow rate density [$J/m^2/s$] |
| $R$ | universal gas constant [J/mol/K]/radius |
| $r$ | radius [m] |
| $s$ | plate thickness [m] |
| $T$ | temperature [K] |
| $t$ | time [s] |
| $V$ | volume [$m^3$] |
| $u$ | specific internal energy [J/kg] |
| $v$ | velocity [m/s] |
| $w_0$ | specific pore volume relative to plate compound mass [$m^3/kg_{comp}$] |
| $X$ | specific uptake [kg/kg] |
| $z$ | spatial coordinate adsorbent [m] |

*Greek letters*

| | |
|---|---|
| $\epsilon$ | void fraction/porosity [$m^3/m^3$] |
| $\lambda$ | heat conductivity [W/m/K] |
| $\eta$ | dynamic viscosity [Pa s] |
| $\rho$ | density [$kg/m^3$] |
| $\tau$ | tortousity [m/m] |

*subscripts*

| | |
|---|---|
| *ads* | adsorber/adsorption |
| *c* | condenser |
| *comp* | absolute mass/volume/cross sectional area of compound |
| *cycle* | cycle |
| *des* | desorber |
| *e* | evaporator |
| *eq* | equilibrium |
| *j* | sum/substance index |

*superscripts*

| | |
|---|---|
| *ad* | adsorbate |
| *bond* | bonding |
| *cond* | condensation |
| *const* | constant |
| *eff* | effective |
| *evap* | evaporation |
| *exp* | experiment |
| *ha* | heat additive |
| *hf* | heat transfer fluid |
| *init* | initial |
| *Kn* | Knudsen |
| *LDF* | linear driving force |
| *l* | liquid |
| *pm* | passive mass |
| *rel* | relative |
| *v* | vapor/sorptive |
| *s* | solid/adsorbent |
| *sat* | saturation |

*Index of abbreviations*

| | |
|---|---|
| *ACH* | adsorption chiller |
| *CDC* | carbide derived carbon |
| *COP* | coefficient of performance |
| *ERR* | error |
| *SCP* | specific cooling power |

## Appendix A. Derivation of Distributed 1D Mathematical Model

The derivation is carried out starting with the three-dimensional, fundamental heat- and material-balances. They are reduced to 1D in the direction of the main transport parts (z-direction), the thickness of the CDC-plates, using the heat- and mass-flow densities shown in Figure A1.

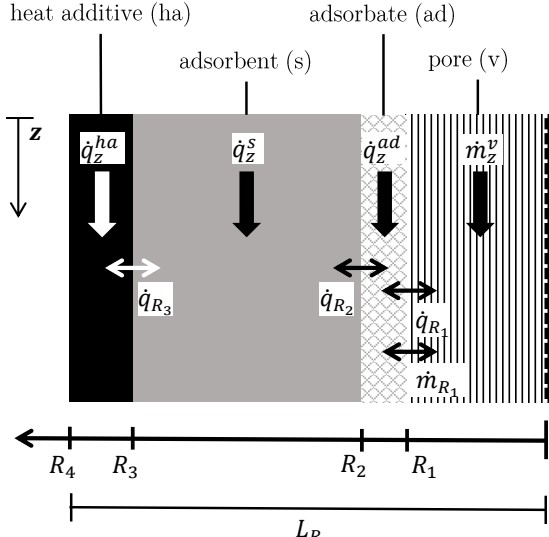

**Figure A1.** Sketch of linear material model with macropore, adsorbate, adsorbent and heat additive. Mass and heat transfer fluxes are indicated with arrows.

Further, the following assumptions are made and relationships apply:

- $\varepsilon^v = \frac{V^v}{V_{comp}}; \quad \varepsilon^s = \frac{V^s}{V_{comp}}; \quad \varepsilon^{ha} = \frac{V^{ha}}{V_{comp}}$
- Volume of adsorbate is not explicitly modeled. It is assumed, that the volume of adsorbent includes solid CDC as well as empty and with adsorbate filled micropores. Therefore, $V^s \approx V^s + V^{ad}$. The mass of adsorbate is related to the total mass of the plate compound giving the specific uptake $X = M^{ad}/M^{ha+s} = M^{ad}/M_{comp}$.
- $\partial V^{ad}/\partial t = \partial V^v/\partial t \approx 0$
- Transport of adsorbate from one CDC-primary particle to another is neglected (no surface-diffusion). Therefore, transport of methanol in z-direction does only occur in the macropores.
- Thermal and material isolation on the outer boundary.

*Appendix A.1. Fundamental Derivation of Quasi-Homogeneous 1D Material Balance*

$$\text{In general, 3D}: \quad \frac{\partial \varrho_j}{\partial t} + (\nabla \cdot \varrho_j \vec{v}) + (\nabla \cdot \vec{j}_j) = \frac{\partial \varrho_j}{\partial t} + \nabla \cdot \vec{\dot{m}}_j = 0$$

Appendix A.1.1. Phase Boundary Conditions Material Balance

$$\left\langle \left[ \left( \varrho_j \vec{v} + \vec{j}_j \right)^+ - \left( \varrho_j \vec{v} + \vec{j}_j \right)^- \right] \right\rangle \cdot \vec{n} = \left\langle \left[ \vec{\dot{m}}_j^+ - \vec{\dot{m}}_j^- \right] \right\rangle \cdot \vec{n} = 0$$

- Macropores (adsorptive)-micropores (adsorbate): At $R_1$

$$\dot{m}_{j,R_1}^v = \dot{m}_{j,R_1}^{ad} \tag{A1}$$

Appendix A.1.2. Material Balance Micropores (Adsorbate)

$$(R_2 - R_1)\frac{\partial \varrho_j^{ad}}{\partial t} - \dot{m}_{j,R_1}^{ad} = 0$$

$$\varepsilon^{ad}\frac{\partial \varrho_j^{ad}}{\partial t} - \frac{\dot{m}_{j,R_1}^{ad}}{L_R} = 0$$

With $\partial V^{ad}/\partial t = \partial V^v/\partial t \approx 0$ and $\varrho_j^{ad}\varepsilon^{ad} = \frac{X_j M_{comp}}{V^{ad}}\varepsilon^{ad} = \frac{X_j M_{comp}}{V_{comp}} = \varrho^{bulk}X_j$

$$\frac{\dot{m}_{j,R_1}^{ad}}{L_R} = \varrho^{bulk}\frac{X_j}{\partial t} \tag{A2}$$

$M_{comp}$ is the overall mass of adsorbent and heat additive and thus, total, dry mass of CDC-plates. $\dot{m}_{j,R_1}^{ad}$ $[kg/m^{2,ad}s]$ is the exchange flux between adsorbate and sorptive and describes the adsorption/desorption mass flux density. Thus, the mass flow per exchange area between adsorbate and sorptive. This can be interpreted as cross sectional area of micropores.

Appendix A.1.3. Material Balance Macropores (Adsorptive Vapour Phase)

$$(R_1 - 0)\frac{\partial \varrho_j^v}{\partial t} + (R_1 - 0)\frac{\partial \dot{m}_{j,z}^v}{\partial z} + \dot{m}_{j,R_1}^v = 0$$

$$\varepsilon^v\frac{\partial \varrho_j^v}{\partial t} + \varepsilon^v\frac{\partial \dot{m}_{j,z}^v}{\partial z} + \frac{\dot{m}_{j,R_1}^v}{L_R} = 0$$

With adsorbate balance (A2) and phase boundary condition (A1):

$$\varepsilon^v\frac{\partial \varrho_j^v}{\partial t} + \varepsilon^v\frac{\partial \dot{m}_{j,z}^v}{\partial z} + \varrho^{bulk}\frac{X_j}{\partial t} = 0 \quad\Bigg|\ \sum_j$$

$$\boxed{\varepsilon^v\frac{\partial \varrho^v}{\partial t} + \varrho^{bulk}\frac{\partial X}{\partial t} + \varepsilon^v\frac{\partial \dot{m}_z^v}{\partial z} = 0} \quad in\ \left[kg/m_{comp}^3 s\right]$$

With $\dot{m}_z^v = -\frac{D^{\text{eff}}}{\varepsilon^v}\frac{\partial \varrho^v}{\partial z}$ (see Equation (7)). Alternatively, with $\dot{m}_z = -D^{\text{eff}}\frac{\partial \varrho^v}{\partial z}$

$$\boxed{\varepsilon^v\frac{\partial \varrho^v}{\partial t} + \varrho^{bulk}\frac{\partial X}{\partial t} + \frac{\partial \dot{m}_z}{\partial z} = 0} \quad in\ \left[kg/m_{comp}^3 s\right]$$

*Appendix A.2. Fundamental Derivation of Quasi-Homogeneous 1D Energy Balance*

$$\text{In general, 3D}: \ \frac{\partial \varrho h}{\partial t} - \frac{\partial p}{\partial t} + \nabla \cdot (\vec{q} + \vec{m}h) - \vec{v}\cdot\nabla p = 0$$

Appendix A.2.1. Phase Boundary Conditions Energy Balance

$$0 = \left[ \left( \varrho h \vec{v} + \vec{q} + \sum \vec{j}_j h_j - \vec{v} p \right)^+ - \left( \varrho h \vec{v} + \vec{q} + \sum \vec{j}_j h_j - \vec{v} p \right)^- \right] \cdot \vec{n}$$

$$0 = \left[ \left( \vec{m} h + \vec{q} - \vec{v} p \right)^+ - \left( \vec{m} h + \vec{q} - \vec{v} p \right)^- \right] \cdot \vec{n}$$

- Heat additive-adsorbent: At $R_3$:

$$\dot{q}^s_{R_3} = \dot{q}^{ad}_{R_3} \tag{A3}$$

- Adsorbate-adsorbent: At $R_2$:

$$\dot{q}^s_{R_2} = \dot{q}^{ad}_{R_2} \tag{A4}$$

- Adsorptive-adsorbate: At $R_1$:

$$\left( \dot{m}_{R_1} h \right)^{ad} + \dot{q}^{ad}_{R_1} - \left( v_{R_1} p \right)^{ad} = \left( \dot{m}_{R_1} h \right)^v + \dot{q}^v_{R_1} - \left( v_{R_1} p \right)^v \tag{A5}$$

Appendix A.2.2. Energy Balance Heat Additive

$$(R_4 - R_3) \varrho^{ha} \frac{\partial h^{ha}}{\partial t} + (R_4 - R_3) \frac{\partial \dot{q}^{ha}_z}{\partial z} - \dot{q}^{ha}_{R_3} = 0$$

$$\varepsilon^{ha} \varrho^{ha} \frac{\partial h^{ha}}{\partial t} + \varepsilon^{ha} \frac{\partial \dot{q}^{ha}_z}{\partial z} - \frac{\dot{q}^{ha}_{R_3}}{L_R} = 0 \tag{A6}$$

Appendix A.2.3. Energy Balance Adsorbent

$$(R_3 - R_2) \varrho^s \frac{\partial h^s}{\partial t} + (R_3 - R_2) \frac{\partial \dot{q}^s_z}{\partial z} + \dot{q}^s_{R_3} - \dot{q}^s_{R_2} = 0$$

$$\varepsilon^s \varrho^s \frac{\partial h^s}{\partial t} + \varepsilon^s \frac{\partial \dot{q}^s_z}{\partial z} + \frac{\dot{q}^s_{R_3} - \dot{q}^s_{R_2}}{L_R} = 0 \tag{A7}$$

Appendix A.2.4. Energy Balance Adsorbate

$$(R_2 - R_1) \left( \varrho^{ad} c^{ad}_p \frac{\partial T^{ad}}{\partial t} + h^{ad} \frac{\partial \varrho^{ad}}{\partial t} + \frac{\partial \dot{q}^{ad}_z}{\partial z} \right) + \dot{q}^{ad}_{R_2} - \dot{q}^{ad}_{R_1} - \dot{m}^{ad}_{R_1} h^{ad} - v^{ad}_{R_1} p^{ad} = 0$$

$$\varepsilon^{ad} \left( \varrho^{ad} c^{ad}_p \frac{\partial T^{ad}}{\partial t} + h^{ad} \frac{\partial \varrho^{ad}}{\partial t} + \frac{\partial \dot{q}^{ad}_z}{\partial z} \right) + \frac{\dot{q}^{ad}_{R_2} - \dot{q}^{ad}_{R_1} - \dot{m}^{ad}_{R_1} h^{ad} - v^{ad}_{R_1} p^{ad}}{L_R} = 0$$

With $\varrho^{ad}\varepsilon^{ad} = \frac{XM_{comp}}{V^{ad}}\varepsilon^{ad} = \frac{XM_{comp}}{V_{comp}} = \varrho^{bulk}X$

$$X\varrho^{bulk}c_p^{ad}\frac{\partial T^{ad}}{\partial t} + h^{ad}\varrho^{bulk}\frac{\partial X}{\partial t} + \varepsilon^{ad}\frac{\partial \dot{q}_z^{ad}}{\partial z} + \frac{\dot{q}_{R_2}^{ad} - \dot{q}_{R_1}^{ad} - \dot{m}_{R_1}^{ad}h^{ad} - v_{R_1}^{ad}p^{ad}}{L_R} = 0 \tag{A8}$$

Appendix A.2.5. Energy Balance Adsorptive

$$R_1\left(\frac{\partial \varrho^v h^v}{\partial t} - \frac{\partial p^v}{\partial t} + \frac{\partial (\dot{m}_z h)^v}{\partial z} - v_z^v\frac{\partial p^v}{\partial z}\right) + \dot{m}_{R_1}^v h^v + \dot{q}_{R_1}^v - v_{R_1}^v p^v = 0$$

With $p^v = \varrho^v T^v R/MW$ and $dh^v = c_p^v dT^v$ for ideal gases

$$R_1\left[\left(\varrho^v c_p^v - R/MW\varrho^v\right)\frac{\partial T^v}{\partial t} + \left(h^v - R/MWT^v\right)\frac{\partial \varrho^v}{\partial t} + \dot{m}_z^v c_p^v\frac{\partial T^v}{\partial z} + h^v\frac{\partial \dot{m}_z^v}{\partial z} - v_z^v\frac{\partial p^v}{\partial z}\right]$$
$$+ \dot{m}_{R_1}^v h^v + \dot{q}_{R_1}^v - v_{R_1}^v p^v = 0$$

$$\varepsilon^v\left[\left(\varrho^v c_p^v - R/MW\varrho^v\right)\frac{\partial T^v}{\partial t} + \left(h^v - R/MWT^v\right)\frac{\partial \varrho^v}{\partial t} + \dot{m}_z^v c_p^v\frac{\partial T^v}{\partial z} + h^v\frac{\partial \dot{m}_z^v}{\partial z} - v_z^v\frac{\partial p^v}{\partial z}\right]$$
$$+ \frac{\dot{m}_{R_1}^v h^v + \dot{q}_{R_1}^v - v_{R_1}^v p^v}{L_R} = 0$$

Using the formulation of the adsorption enthalpy (19) and with the energy balances of heat additive (A6), adsorbent (A7) and adsorbate (A8) and the phase boundary conditions (A3), (A4) and (A5), the quasi-homogeneous 1D energy balances follow

$$\boxed{\begin{aligned}&\left(\varrho^{bulk}Xc_p^{ad} + \varepsilon^{ha}\varrho^{ha}c_p^{ha} + \varepsilon^s\varrho^s c_p^s + \varepsilon^v\varrho^v(c_p^v - R/MW)\right)\frac{\partial T}{\partial t} - \varepsilon^v R/MWT\frac{\partial \varrho^v}{\partial t}\\ &+ \varepsilon^v\dot{m}_z^v c_p^v\frac{\partial T}{\partial z} - \varepsilon^v v_z^v\frac{\partial p^v}{\partial z} + (1-\varepsilon^v)\frac{\partial \dot{q}_z^{ha+s+ad}}{\partial z} + \Delta_{ads}h\varrho^{bulk}\frac{\partial X}{\partial t} = 0\end{aligned}}\quad in\ \left[J/m_{comp}^3 s\right]$$

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
