# Peer review of "Simultaneous Optimization of Process Operational and Material Parameters for a 2-Bed Adsorption Refrigeration Process"

_2305-7084, doi:10.3390/chemengineering4020031_

Round 1

Reviewer 1 Report

Accept in present form.

Author Response

Dear Sir or Madame,

thank you for your review.

We have revised the manuscript and made some minor changes based on the reviewer’s suggestions. The changes are highlighted in the newly submitted manuscript.

Best regards,

Marc Scherle

Reviewer 2 Report

The issue discussed  in this work is very interesting and promising. This submission is a well written piece of theoretical research which is worth publishing.

1: Are these approaches the subject of future work by your research team?

2: Please, indicate clearly in the text the advantages of the method considered in relation to the others.

3:Could the authors comment on the reliability of the theoretical model?

Author Response

Dear Sir or Madame,

thank you for your review and your suggestions.

Based on your language evaluation, we have revised the manuscript and made some English formulation and grammar changes. The changes are highlighted in the newly submitted manuscript.

Regarding your questions:

  1. Yes, these approaches are subject of future work. First, we plan to stick to adsorption refrigeration as a model process and to apply the method to other process setups with more sophisticated material models. Our goal is to show the benefit of material design, 3D-printing, … in the field of process engineering.
    Subsequently, we plan to extend the method and apply its general concept on other, more difficult processes in process engineering.
  2. The following lines have been added to the manuscript.
    Taking both, SCP and COP, into account is advantageous since the user can decide whether a high power density or efficiency is the main goal.
    Even though the presented method is demanding in terms of calculation effort, it offers some advantages compared to other methods. Using a one-dimensional model with effective transport parameters is advantageous. Once a material is parameterized, a variation of the layer thickness is possible without the need for new experiments and parameterizations. In comparison, zero-dimensional models usually only using one kinetic parameter, must be parameterized for each new layer thickness. This requires a high effort in material production and experimental characterization. Once a valid material model has been found, the method can be applied to a variety of different process setups. Another advantage of this method is its flexibility. It can be easily extended to more sophisticated material models and applied to several different process setups.
    Each combination of material and process results in a unique optimal curve. Visualizing the final optimization result in a master curve that contains all relevant material and process related optimization is another advantage of this method, because it allows for an easy comparison of different process arrangements and material models.
  3. The theoretical model provides an insight in specific uptake and temperature profiles in the consolidated adsorbents, which are experimentally not accessible. However, the pressure and temperature on the material boundaries are measurable. To measure adsorption capacity, we’ve used static-volumetric adsorption measurements. Besides adsorption capacity in equilibrium, these experiments provide pressure and temperature distributions over time from beginning of the experiments to equilibrium. These kinetical profiles could be reproduced very well with our theoretical model.  Nevertheless, these experiments are not of cyclic nature and in comparison to the adsorption refrigeration process different boundary conditions are applied. Therefore, a validation of the theoretical model to cycle adsorption experiments is pending. To tackle this issue, we’ve developed and constructed an experimental setup with one adsorber module, which offers the opportunity of conducting cyclic adsorption experiments. First comparisons of experiments and simulations were quite promising. We plan to publish these works later this year.
    Concluding, the theoretical model is reliable if it comes to dynamic simulations of heat- and mass-transfer and adsorption/ desorption in consolidated adsorbents but a complete validation of the process model is pending.

Best regards,
Marc Scherle

Reviewer 3 Report

The Manuscript ChemEngineering-756010 titled "Simultaneous optimization of process operational and material parameters for a 2-bed adsorption refrigeration process" presents a study that combines the influence of both the structural materials properties of the adsorption beds, and the operational process parameters (i.e, cycle time, adsorbent thickness) during a two-bed adsorption refrigeration process.

The authors introduced the topic well, highlighting the importance of increasing the efficiency of this type of systems given their environmentally friendly energy requirements. They mentioned the importance of using novel composite porous materials (carbide-derived carbon) and thermally conductive additives (boron nitride platelets) to improve the thermal conductivity of the material while maintaining permeability for an effective heat and mass transfer. In addition, and parallel to this, they emphasize the importance of process optimization for which they run a series of modeling capabilities for the adsorption process in study. Out of this modeling study, the authors determined optimal bed thicknesses and cycle times with the intention of reaching a global optimum.

I believe this work has achieved what it meant, as it addressed the gap between material development and process optimization. Both, the experimental and modeling work are relevant and deserved publication. I think though, that more could have been done on the material side of the research as other more conductive additives could have been included in the study also. However, given the extension and the scope of the work, this is enough for a single publication.

The authors present in great detail the experimental procedures, and the modeling is undertaken with an advanced mathematical level and sound transport phenomena basis. Thus, experimentally and theoretically I find nothing wrong.

The English language utilized is at a very good level, though some little typos are still found. Yet, no problem at all for the clarity and the overall message.

The conclusions back up the results; however, the whole paper is mainly dedicated to the modeling aspects of the study and little is related to the materials used and performance.

Overall, it is an informative and original work useful for the scientific and technical community. I recommend it for publication in this journal.

Author Response

Dear Sir or Madame,

thank you very much for your detailed review and feedback. This is very helpful for us.

As you lay out, this paper is mainly dedicated to show our methodology and the simulation works, which in our opinion is the main new value for the community. To clarify our approach and focus on the methodology, we’ve decided to use a very simple material model and only briefly describe the experimental work. Nevertheless, the methodology is easily extendable to more sophisticated material models, other process setups and even completely different processes in the field of process engineering.
Furthermore, a detailed work on the materials fundamentals is planned and should soon be published by our project partner Prof. Etzold who provided the carbide-derived carbon samples.

Based on your language suggestion, we have revised the manuscript and made some English formulation and grammar changes. The changes are highlighted in the newly submitted manuscript.

Best regards,

Marc Scherle